# Enhancing the Anticancer Efficacy of Immunotherapy through Combination with Histone Modification Inhibitors

**DOI:** 10.3390/genes9120633

**Published:** 2018-12-14

**Authors:** Wanyu Sun, Shuting Lv, Hong Li, Wei Cui, Lihui Wang

**Affiliations:** Department of Pharmacology, Shenyang Pharmaceutical University, Shenyang 110016, China; m18340467855_1@163.com (W.S.); lstbmyy@163.com (S.L.); Lh_08250517@163.com (H.L.)

**Keywords:** histone methylation, acetylation, immunotherapy, cancer

## Abstract

In the nucleus of each cell, the DNA is wrapped around histone octamers, forming the so-called “nucleosomal core particles”. The histones undergo various modifications that influence chromatin structure and function, including methylation, acetylation, ubiquitination, phosphorylation, and SUMOylation. These modifications, known as epigenetic modifications (defined as heritable molecular determinants of phenotype that are independent of the DNA sequence), result in alterations of gene expression and changes in cell behavior. Recent work has shown that epigenetic drugs targeting histone deacetylation or methylation modulate the immune response and overcome acquired resistance to immunotherapy. A number of combination therapies involving immunotherapy and epigenetic drugs, which target histone deacetylation or methylation, are currently under various clinical/pre-clinical investigations and have shown promising anticancer efficacy. These combination therapies may provide a new strategy for achieving sustained anticancer efficacy and overcoming resistance.

## 1. Introduction

Within the cell nucleus, DNA is organized into “nucleosomal core particles”, which consist of about 145–147 bp of DNA wrapped around a histone octamer [1]. Each histone octamer contains two of each of the core histone proteins (H2A, H2B, H3, H4) [2,3]. Post-translational modifications of histones can occur throughout the protein, but modifications occurring in the N-terminal tails have been most thoroughly investigated. The N-terminal regions of the histone tails protrude from the nucleosome and can easily undergo various modifications, including methylation and acetylation [1,4,5]. These modifications influence the structure of chromatin and affect the accessibility of DNA for transcription factors and other proteins, thus activating or inactivating gene expression and finally regulating the cell’s behavior. Histone modifications are known as epigenetic changes, because they affect gene expression without changing the DNA sequence. The dysregulation of histone modification correlates with the development of disease, especially cancers [6]. However, the mechanism of histone modification and the role of histone dysregulation in disease development are not fully understood. Some agents, such as Vorinostat (suberanilohydroxamic acid, SAHA) and Belinostat, which target the enzymes that regulate histone post-translational modification, have been approved by the FDA for use in clinical cancer treatment. These agents have shown good efficacy when used in the clinical treatment of cancers. However, there are limits to their application, including the non-responsiveness or relapse of some patients, the lack of predictable therapeutic indicators, and the existence of associated adverse events [7,8]. Recently, some studies have shown that post-translational modification of histones may regulate the behavior of cancer cells and cells involved in the immune response, including regulatory T cells (Tregs), dendritic cells (DCs), myeloid-derived suppressor cells (MDSCs), effector T cells (Teffs), and others [9]. Histone modification inhibitors may increase the immunogenicity of tumor cells, hence enhancing their recognition and elimination by the immune system [10]. These results imply that agents targeting histone modification, when combined with immunotherapy, may enhance the therapeutic efficacy. This line of reasoning has triggered some pre-clinical/clinical investigations of combination strategies involving histone modification inhibitors and immune approaches, such as immune checkpoint blockade therapy and adoptive cellular immunotherapy or cytokine-based therapy.

## 2. Histone Modifications

When the chromatin structure is highly compact, the efficient recruitment and processivity of RNA Polymerase II (Pol Il) is impeded and transcription is thus hindered [11]. The dysregulation of histone post-translational modification alters the chromatin structure and induces abnormal gene expression, which is correlated with the occurrence and development of diseases, especially cancers [6]. Modifications of all four histones have been reported, but those affecting H3 and H4 are best understood [12]. Among these histone modifications, the regulatory role and function of acetylation and methylation have been studied most intensively.

### 2.1. Histone Acetylation Modifiers

Histone acetylation is an important modification that mainly occurs in the N-terminal region of histone tails. This modification is a mark of active chromatin, and usually weakens the binding of histones to DNA, thus “relaxing” the chromatin and enhancing gene expression, as shown in Figure 1 [13]. The addition of acetyl groups to histones is regulated by histone acetyl transferases (HATs), while removal of acetyl groups is catalyzed by histone deacetylases (HDACs) [14]. The HDACs are divided into four classes. Class I HDACs include HDAC1, HDAC2, HDAC3, HDAC8; class II HDACs are further divided into two groups, class IIa (HDAC4, HDAC5, HDAC7, HDAC9) and class IIb (HDAC6, HDAC10); class III contains SIRT1-7; and class IV contains one enzyme, HDAC11 [15,16]. Classes I, II, and IV HDACs are all Zn^2+^-dependent enzymes, while class III HDACs do not show any sequence similarity to the other three classes and depend on NAD^+^ as a co-factor [15,16]. By reversing the histone acetylation status, HDACs mostly inhibit gene expression (Figure 1). The dysregulation of HDACs is correlated with the occurrence of various diseases, including cancer [14]. 

Because of the clear correlation between histone deacetylation and tumor development, HDAC inhibitors (HDACis) have been investigated in clinical trials for cancer therapy [14]. The results indicate that inhibition of HDACs induces the growth arrest and apoptosis of tumor cells [6] and enhances immunity [7]. Several HDACis investigated in clinical trials showed anti-tumor effects and have been approved for the clinical treatment of various types of cancer [17]. For example, Vorinostat was approved for the treatment of cutaneous T-cell lymphoma (CTCL) in 2006 [18]. Belinostat, which shows efficacy for the treatment of peripheral T-cell lymphoma (PTCL), was approved in 2014. Panobinostat (LBH-589) was approved for treatment of multiple myeloma (MM) in 2015 [13,18]. The use of Romidepsin in CTCL and PTCL therapy was approved in 2009 and 2010, respectively [15]. Also, Chidamide was approved in China in 2014 for treatment of PTCL.

As mentioned above, HATs are another important class of enzyme that regulate histone acetylation. They catalyze the addition of acetyl groups to residues in histone tails or histone core regains using acetyl coenzyme A (acetyl-coA) as the donor [12]. Acetylation weakens the binding between histone and DNA, which induces a relaxed chromatin status and promotes gene expression [12]. HATs have not been investigated as intensively as HDACs. On the basis of their structure and the functional properties of their catalytic centers, HATs can mainly be grouped into five categories: Gcn5-related acetyltransferases (GNATs); the MOZ, Ybf2/Sas3, Sas2, Tip60 (MYST)-related HAT superfamily; p300/CBP HATs; TAF250 HATs; and steroid receptor co-activator and nuclear receptor co-activator family (NCoA) HATs [19]. HATs located in different complexes may catalyze the modification of different residues. HATs also catalyze the acetylation of non-histone protein substrates and affect their activity, including a number of transcription factors [5]. By dysregulating the acetylation status of histones and other non-histone proteins, HATs play an important role in oncogenesis [20]. For example, MOF, a member of the MYST family which preferentially catalyzes acetylation of lysine 16 in H3 (H3K16), shows aberrant expression in some cancer phenotypes such as breast cancer, medulloblastoma, ovarian cancer, renal cell carcinoma, and non-small cell lung cancer (NSCLC) [20]. In addition, NCOA3 (AIB1), a member of the NCoA HAT family, was reported to be highly expressed in breast cancer cells, and in certain leukemias. Unlike MOF and NCOA3, Tip60, a member of MYST HAT family, plays an anti-tumor role [21]. Decreased expression of Tip60 induces hypoacetylation of the p53 protein, hence suppressing apoptotic signals [5]. This result shows that Tip60 may act as a tumor suppressor protein. Mono-allelic loss of the *Tip60* gene was observed in lymphomas, mammary carcinomas, and head and neck tumors [21]. In addition to the above mentioned histone deacetylation inhibitor, recently some inhibitors targeting HATs have also been investigated for cancer therapy and shown prospective efficacy. It has been reported that A-485 as a potential selective inhibitor of p300 and CBP showed selectively inhibitory proliferation in lineage-specific tumor types, including several hematological malignancies and androgen receptor-positive prostate cancers [22], suggesting the potential of targeting HAT in cancer treatment.

The bromodomain and extra-terminal (BET) family proteins, as “readers” of chromatin status, can recognize and bind to acetyl-lysine residues [13]. They subsequently interact with and recruit positive transcription elongation factors (pTEFb) and mediator complexes to chromatin [18,23]. BET family proteins were reported to participate in the development of various cancers, including acute myeloid leukemia (AML), myelodysplastic syndrome (MDS), lymphoma, glioblastoma, NUT midline carcinoma (NMC), multiple myeloma, breast cancer, NSCLC, and prostate cancer. In addition, small molecule inhibitors of BET family proteins, such as OTX015 and CPI-0610 [23], are currently under investigation in different phases of clinical trials [13]. The structure of representative histone acetylation modulators are shown in Table 1.

### 2.2. Histone Methylation Modifiers

In contrast to histone acetylation, histone methylation regulates gene expression depending on the modified residues and the number of methyl groups, as shown in Figure 2 [13]. Methylation status is regulated by histone methyltransferases and demethylases. More than 60 kinds of histone methyltransferase (HMT) and demethylase (HDM) have been identified [24]. HMTs are divided into those that methylate histone lysine residues (HKMTs) and those that methylate histone arginine residues (PRMTs). Histone arginine residues can be mono-methylated or di-methylated, and histone lysine residues can be mono-methylated, di-methylated, or tri-methylated [1]. HMT and the related complex catalyze the mono-methylation, di-methylation, or tri-methylation of specific residues and play differential roles in the regulation of gene expression. Methylated residues are recognized by several important protein domains, including the chromo-domain, tudor domain, and MBT domain. Proteins containing these domains can bind the methylated residues and recruit various chromatin remodeling complexes, hence regulating the chromatin structure and gene expression. Among the various HMT enzymes, EZH2 and DOT1L, which both belong to the PKMT family, were shown to be highly expressed in tumors [25]. Mutations in the genes encoding isocitrate dehydrogenase (IDH) have been observed in some types of cancer [20]. The mutant proteins are able to suppress the activity of histone demethylases and hence induce histone hypomethylation.

The most intensively studied residues that undergo methylation are lysines 9 and 27 of H3 (H3K9 and H3K27). Methylation of H3K9 and H3K27 results in suppression of gene expression [5,26]. In contrast, methylation of H3K4 and H3K36 will relax the chromatin structure and promote gene transcription, as shown in the Figure 2 [5,26]. The lysine methyltransferase EZH2, as the catalytic sub-unit of polycomb repressive complex 2 (PRC2), mediates the mono-, di-, and tri-methylation of H3K27 (H3K27me1/2/3) [18,25]. EZH2-catalyzed tri-methylation of H3K27 induces the suppression of gene expression and is correlated with the onset and development of various types of cancer [26]. Another well-studied HMT, DOT1L, catalyzes the methylation of H3K79, which promotes transcription [18]. H3K79me3 is often observed at the transcriptional start sites of active genes [27]. DOT1L has been shown to play an important role in the development of KMT2A-rearranged AML [13].

Histone demethylases are another family of enzymes that are important for regulating histone methylation. HDMs are “erasers” of methylation, and their dysregulation is related to tumor biology [13]. LSD1, the first HDM to be discovered, catalyzes the demethylation of H3K9 and H3K4 [18]. Many other HDMs have since been identified, and they are divided into two groups: the Jumonji AT-rich interactive domain 1 (JARID1) family, which catalyzes a FAD-dependent reaction, and the Jumonji C (JmjC) domain-containing protein (JMD2) family, which remove methyl groups using α-ketoglutarate, Fe^2+^, and molecular oxygen as cofactors [19,20]. Like the HMTs EZH2 and DOT1L, the HDM LSD1 is also correlated with tumor onset and progression, and some inhibitors of LSD1, including GSK2879552 and INCB059872, are under investigation in different phases of pre-clinical and clinical trials for application in the treatment of AML, MDS, and NSCLC [13]. 

Restoration or inhibition of the activity of histone methyltransferases is of great importance for tumor therapy. Some HMT inhibitors (HMTis) are under investigation in different phases of clinical trials. They include Pinometostat (EPZ-5676), an inhibitor of DOT1L, and Tazemetostat (EPZ-6438), an inhibitor of EZH2, which are being investigated for the treatment of AML, MM, and MDS [25]. Tazemetostat is also under phase II clinical trials in adults with INI1-negative tumors or relapsed or refractory synovial sarcoma (NCT02601950). Pinometostat, which targets DOT1L, was investigated in phase I trials in adults with relapsed or refractory MLL-r leukemia (NCT01684150), and in pediatric patients with relapsed of refractory MLL-r leukemia (NCT02141828). The structure of representative histone methylation modulators are shown in Table 1.

### 2.3. Histone Phosphorylation and Ubiquitination Modifiers

The phosphorylation of histones is catalyzed by specific kinases, which add phosphate groups to tyrosine, threonine, and serine residues [28]. Phosphorylation of H3 at serine 10 causes dissociation of the heterochromatin packing proteins HP1α, HP1β, and HP1γ from nucleosomes, allowing the access of factors needed for proper condensation and segregation of the chromosomes [28]. CK2 and DNA-PK phosphorylate H4 (at S1) and the linker histone H1 (at S18, S173, S189, T11, T138, and T155) [28], respectively, but the role and function of these modifications are still not known.

Ubiquitin is a highly conserved eukaryotic protein of about 76 amino acids in length. Ubiquitination refers to the covalent attachment of one (monoubiquitination) or more (polyubiquitination) ubiquitin monomers to the amino group of a lysine residue [28]. Polyubiquitination is catalyzed by three enzymes: a ubiquitin-activating enzyme (E1), a ubiquitin-conjugating enzyme (E2), and a ubiquitin-protein ligase (E3) [28]. Monoubiquitination requires only E1 and E2. Polyubiquitination marks a protein for degradation. The function of these monoubiquitination and polyubiquitination events depends on the residue that is modified. Ubiquitination mainly occurs on H2A and H2B, and is less frequent on H3, H4, and H1 [28]. Although H2A and H2B may be polyubiquitinated, the monoubiquitinated forms are by far the most abundant. Monoubiquitination of H2A is correlated with the inactivation of X chromatin and affects histone methylation, hence suppressing gene transcription [28]. Histone phosphorylation and ubiquitination are less well investigated than acetylation and methylation, but they can modulate chromatin structure and gene expression by influencing acetylation and methylation.

## 3. Modulation of the Immune Response by Histone Modification 

### 3.1. The Role of the Immune System in Tumorigenesis and Tumor Cell Escape

The immune system, which consists of innate and adaptive host immune cells and other signaling components, is able to recognize and eliminate abnormal cells [29]. Cancer cells expressing tumor-associated antigens (TAAs) on the surface can be recognized and eliminated by natural killer (NK) cells and T cells [7]. Tumor cells with lower immunogenicity cannot be detected by the immune system; this is called “immune escape” and it promotes tumor formation and progression [10]. The mechanisms underlying immune escape mainly include anti-apoptotic signaling, mitogen-activated protein kinase (MAPK) signaling, microphthalmia-associated transcription factor (MITF), cyclic adenosyl monophosphate (cAMP), and NF-κB [10]. These mechanisms reduce the ability of the immune system to detect and destroy the tumorigenic cells. Suppression of immune function by the tumor microenvironment (TME) is another important mechanism underlying immune escape [30]. The TME suppresses the immune response by downregulating antigen presentation by tumor cells, promoting tumor-induced immune checkpoint suppression, enhancing the functions of Treg cells and MDSCs, and suppressing the secretion of immunosuppressive cytokines [31]. These effects create a dynamic process termed “immunoediting”, which includes three phases: elimination, equilibrium, and escape [10]. Tumor cells with lower immunogenicity are selected in the elimination phase, and they then differentiate or proliferate during the equilibrium phase, thus promoting tumor progression. Considering these factors, there is a great need to investigate agents that target tumor cells in the different immunoediting phases. 

### 3.2. Effect of Histone Modification on the Immune Response

As discussed above, the dysregulation of histone modification correlates with the onset and development of tumors. Some agents designed to target enzymes or other proteins that participate in histone modification have been tested in pre-clinical and clinical trials and show therapeutic efficacy in the treatment of tumors. Some of these agents have been approved by the FDA for clinical application. Although these approved agents have shown potential in trials, there are limitations to their use in the clinic, because some patients do not respond to therapy or relapse after treatment. However, recent research has shown that histone modifications may regulate the immune response to some degree as well as affecting the immunogenicity of cancer cells [7]. These results suggest that histone modification inhibitors, in combination with immunotherapy agents, may deliver enhanced therapeutic efficacy.

#### 3.2.1. Modulation of the Immune Response by Histone Acetylation

(1) Histone acetylation modulators regulate the immunogenicity of cancer cells

Histone acetylation affects chromatin structure and may regulate the immune response by modulating the expression of various genes related to the immune system. Hence, it is proposed that inhibitors of enzymes that control the histone acetylation status may modulate the immune response. It has been reported that HDACis increase the immunogenicity of cancer cells by upregulating the expression of TAAs, components of the antigen processing and presentation machinery (APM), co-stimulatory molecules, stress-induced ligands and death-inducing receptors, and down-regulating the expression of checkpoint ligands by tumor cells [32]. Cancer testis antigens (CTAs), which are only expressed in placenta and testis and are silenced in mature somatic cells, are regarded as the best characterized class of epigenetically-regulated TAAs [32]. The repression of CTA expression was observed in some tumor phenotypes and may downregulate the immunogenicity of the cells, thus allowing them to escape from immune surveillance [33]. HDACis such as Valproic acid (VPA) and Trichostatin A (TSA) upregulate the expression of CTAs; however, they may not upregulate the expression of all kinds of TAA [29,32]. Apart from CTAs, HDACis also upregulate the expression of APM components including major histocompatibility complex (MHC) molecules, TAP-1, TAP-2, LMP-2, and LMP7 [29]. In addition, several different HDACis, including Dacinostat, were also shown to upregulate the surface expression of co-stimulatory molecules including CD40, CD80, CD86, and ICAM-1, the stress-induced ligands MICA and MICB, and the death-inducing receptor FAS and TRAIL receptors [32]. The HDACi Panobinostat also enhanced the expression of immune checkpoint ligands on the surface of tumor cells, including PD-L1 and PD-L2, and of checkpoint receptors (CTLA-4, PD-1) on the surface of tumor-infiltrating lymphocytes (TILs), thus affecting the efficacy of immune checkpoint blockade therapy [10]. 

(2) Histone acetylation modulators regulate NK cells

HDACis not only increase the immunogenicity of cancer cells, but also enhance the immune response by regulating the adaptive and innate host immune cells, which recognize and eliminate cancer cells [32]. NK cells are key mediators of the innate immune response. They are able to recognize cancer cells and virus-infected cells and once activated, they are cytotoxic [7]. The activation of NK cells is mediated by the interaction of NKG2D receptors on the surface of NK cells with ligands, including ULBP and the MHC class I chain-associated proteins MICA and MICB, which are expressed on the surface of tumor cells. NK cells kill tumor cells through death receptor signaling and secretion of cytotoxic granules [33]. HDACis are able to enhance the expression of MHC molecules and ULBPs, and hence enhance the ability of NK cells to recognize and eliminate cancer cells [26,32]. NKG2D, the immunoreceptor that binds with MICA and MICB, was upregulated in NK cells by Valproic acid (VPA), and then enhanced the NK-mediated lysis of cancer cells in AML [7,34]. 

(3) Histone acetylation modulators regulate cytotoxic T lymphocytes

Cytotoxic T lymphocytes (CTLs) are key mediators of the adaptive immune response [7,32]. In the TME, CTLs induce tumor cell death through death receptor interaction and the secretion of cytotoxic granules. The activation of CTLs is mediated by the binding of the T cell receptor (TCR) and human leukocyte antigens (HLA) class I molecules expressed on the antigen-presenting cells (APCs), along with a co-stimulatory or accessory signal (the binding of CD80 or CD86 on APCs with CD28 on T cells) [32,35]. HDACis are able to upregulate the expression of HLA class I molecules and co-stimulatory modules on the surface of tumor cells, hence upregulating the killing ability of the CTLs [7,33]. During the differentiation of CTLs from naive cells, transcriptional activation of the gene encoding IFN-γ is related to the upregulation of H3K9 acetylation [9]. HDACis have been shown to upregulate the expression of chemokines that attract T cells, which enhances the infiltration of Teff cells into the TME [32,36]. The downregulation of chemokine expression will promote the immune escape of tumor cells.

(4) Histone acetylation modulators regulate APCs

In the TME, tumor-associated macrophages (TAMs), together with DCs, act as APCs and play important roles in TME-influenced tumor growth and progression [4]. Research has shown that HDACis modulate the function of these two cell types, hence regulating the immune response [37]. There are two TAM phenotypes, the pro-tumorigenic M2 phenotype, and the anti-tumorigenic M1 phenotype [38]. HDACis may regulate the populations of these two cell types [37]. TMP195, a specific inhibitor of Class IIa HDACs, has been shown to recruit circulating monocytes into the tumor, hence decreasing the immunosuppressive M2 population and increasing the anti-tumorigenic M1 population in a mouse model of breast cancer [38]. DC cells, which are also potent APCs, mediate antigen-specific anti-tumor responses and present antigens, mainly by scraping them from the membranes of dying tumor cells and by phagocytosis of the membrane of live cells. DCs hence promote the elimination of tumor cells by CTLs [39]. 

(5) Histone acetylation modulates immunosuppressive cells

Treg cells and MDSCs are two kinds of repressive cell observed in the TME. They are both downregulated by HDACis, and therefore HDACis reduce immune suppression [9]. Entinostat, a pan-inhibitor of HDACs, has been reported to cause eradication of tumors and metastasis in a mouse model of breast cancer, mainly through inhibition of granulocytic MDSCs [11]. The histone acetyltransferase EP300 acetylates histones in the *FOXP3* gene and enhances the stability of FOXP3 by preventing the ubiquitination and subsequent degradation of FOXP3 protein, hence promoting the function of Tregs [29]. Inhibitors of EP300 reverse the suppressor function of Treg cells, repress the induction of cytokines, and suppress the growth of tumors in mice [11]. In addition to these mechanisms, HDACis reduce the population of repressive cells, thus reversing the immune suppression. Additionally, some papers have reported that HDACis are able to suppress the activity of immunosuppressive tumor-endogenous lymphocytes [40]. The effects of HDACis on the immune response are summarized in Figure 3.

#### 3.2.2. Modulation of the Immune Response by Histone Methylation

(1) Histone methylation modulators regulate T cells and NK cells

Similar to histone acetylation, histone methylation can regulate the immune response by modulating the immunogenicity of cancer cells and by affecting the expression of molecules of the immune response [11]. Chemokines have been shown to enhance the infiltration of effector T cells into the TME, hence promoting the killing of tumor cells [32]. The HMT EZH2 modulates the tri-methylation of H3K27 to suppress the expression of the T helper1 (Th1)-type chemokines CXCL9 and CXCL10, which reduces the infiltration of effector T cells in ovarian cancer [32,41]. Selective inhibitors of EZH2 are able to reverse the methylation of H3K27 and upregulate the expression of CXCL9 and CXCL10, thus promoting the infiltration of effector T cells into the TME, and hence achieving therapeutic efficacy in lung cancer [32].

HMTs also modulate other components of the immune response. EZH2 suppresses the differentiation and function of NK cells by downregulating the expression levels of the NKG2D receptor. In addition, EZH2-mediated H3K27me3 is able to induce silencing of the IL-15R, CD122, and NKG2D receptor proteins, hence suppressing NK cell expansion and decreasing the cytotoxic targeting of tumor cells.

(2) Histone methylation modulators regulate APCs

As mentioned above, TAMs play a crucial role in antigen presentation. M1 macrophages are anti-tumorigenic, while M2 macrophages can promote tumor growth. Studies have shown that HMTs modulate the transition of the M1 phenotype to the M2 phenotype [4]. Inhibitors targeting PRMT1 and JMJD3 in TAMs prevent the M1-to-M2 transition, hence suppressing tumor growth [4]. DCs are also important APCs in the immune system, and their differentiation and function are related to the methylation status of H3K27me3 and H3K4me3 [37]. The immunosuppressive molecule TGF-β is able to induce H3K4me3- and H3K27me3-related chromatin remodeling in DCs, thus triggering upregulated transcription of genes encoding co-stimulatory modules and cytokines, and downregulated expression of differentiation markers. 

(3) Histone methylation modulators regulate immunosuppressive cells

EZH2 has also been shown to play an important role in maintaining the function of Treg cells [42]. Accumulation of H3K4me3 in the promoter of the *FOXP3* gene induces transcription of *FOXP3* and results in the generation of Tregs. Recently, a study uncovered that tumor-infiltrating Tregs (TI-Tregs) acquire pro-inflammatory functions through pharmacological or genetic suppression of the activity of EZH2 [42]. Suppression of EZH2 modulates the TME and enhances the infiltration of CD8+ and CD4+ effector T cells, which can eliminate tumors [42].

HMTs and HDMs also modulate other immune-repressive cells. Cancer-associated fibroblasts (CAFs) are stromal cells that play an immunosuppressive role in the TME [26]. Studies have shown that histone demethylation in breast CAFs promotes the expression of pro-tumorigenic genes, thus allowing tumor cells to escape from immune surveillance [9]. The effects of HMTis on the immune response are summarized in Figure 3.

## 4. Histone Modification Inhibitors Enhance Immune Therapy Efficacy

### 4.1. Immune Therapy and Cancer

Currently, immunotherapy is the most exciting and promising field for achieving long-term management of advanced human tumors [43,44]. Therapy with immune checkpoint blockade inhibitors is an especially interesting topic [31,45]. The aim of immunotherapy, defined as enhancing human immune functions, is to kill the cancer cells. Unlike surgery, chemotherapy, and radiotherapy, which target tumor cells or tissue directly, immunotherapy achieves anti-tumor effects mainly by modulating the immune system. There are main four directions of immunotherapy investigation: (1) adoptive cellular immunotherapy; (2) cytokine-based therapy; (3) vaccines; and (4) immune checkpoint inhibitors [32]. Conventional immunotherapy suppresses the growth and development of tumors mainly by “evoking or reinforcing the host immune reactions against the tumor” [35]. It can be difficult to achieve the expected response from this approach because of adverse events associated with overreaction of immune cells in non-tumor organs [35]. In contrast, immune checkpoint blockade therapy suppresses the development of the tumor mainly through “improving the immune response by decreasing the suppression of immune checkpoint inhibitors” [35]. When applied in the clinic, therapies based on immune checkpoint blockade decrease the incidence of unwanted side-effects in healthy organs and are therefore more effective than conventional immunotherapy [10]. Today, pre-clinical investigations of immune checkpoint blockade drugs mainly focus on three immune checkpoint molecules: CTLA-4, PD-L1, and PD-1 [41,45,46]. Four immune checkpoint inhibitors—Ipiliumab (antibody targeting CTLA-4), Nivolumab (antibody targeting PD-1), Pembrolizumab (antibody targeting PD-L1), and Atezolizumab (antibody targeting PD-L1)—have been approved by the FDA to treat locally advanced or metastatic urothelial carcinoma in patients who do not have the expected response to chemotherapy or radiotherapy [47,48]. Tremelimumab, an antibody against CTLA-4, is in phase III investigation. Pidilizumab, an antibody against PD-1, is in phase I/II trial. Atezolizumab, which targets PD-L1, is in phase III/IV trial. Beyond that, antibodies targeting other immune checkpoint molecules, such as LAG-3, OX-40, and CA-47, are being tested in several pre-clinical investigations. However, in clinical trials, some cancer patients do not respond to immune checkpoint blockade therapy or relapse after therapy [49,50]. This may be related to loss of expression of immune components including TAAs, antigen processing and presentation molecules, or co-stimulatory molecules. Notably, some recent studies showed that combining epigenetic therapy drugs with immune checkpoint inhibitors may synergistically enhance the response of cancer patients and decrease the possibility of relapse [32,43]. These findings suggest a promising new way to fight cancer.

### 4.2. HDAC Inhibitors in Combination with Immunotherapies in Pre-Clinical and Clinical Trials

#### 4.2.1. HDAC Inhibitors Combined with Immune Checkpoint Inhibitors in Pre-Clinical and Clinical Trials

As described above, aberrant histone acetylation modification modulates the immune response. This indicates that inhibitors of histone deacetylation may regulate the expression and function of immune system components and provide a new strategy for cancer therapy. Some treatments involving HDAC inhibitors combined with immune therapy have demonstrated promising efficacy in various phases of pre-clinical and clinical trials [43]. Currently, most investigations are focusing on combinations of HDAC inhibitors and immune checkpoint inhibitors for cancer treatment. 

(1) HDAC inhibitors combined with immune checkpoint inhibitors targeting PD-1/PD-L1 in pre-clinical and clinical trials 

At present, the most frequently investigated combination in pre-clinical/clinical trials is an HDACi plus an immune checkpoint inhibitor targeting PD-1/PD-L1. As explained above, histone acetylation modulators enhance the efficacy of immune checkpoint inhibitor therapy by increasing the expression of immune checkpoint ligands and TAAs on tumor cells, which upregulates the production of chemokines by T cells and reduces the population of immunosuppressive cells in the TME [32]. Previous studies have indicated that treatment with an HDACi before an immune checkpoint inhibitor can enhance the efficacy of immunotherapy [32]. Research has uncovered that in melanoma-bearing mice, treatment with HDACi enhanced the acetylation level and increased the expression of the immune checkpoint ligands PD-L1 and PD-L2 on the tumor surface [32]. Increased expression of immune checkpoint molecules on tumor cells after HDACi treatment improved the response to immune checkpoint blockade therapy [32]. Hence, this treatment strategy repressed tumor growth and prolonged overall survival. HDAC inhibitors may also enhance immune checkpoint blockade therapy through other signaling mechanisms. It has been reported that in ovarian tumor-bearing mice, an HDACi plus the immune checkpoint inhibitor anti-PD-1 activated type I IFN signaling and then reduced immunosuppression and the tumor burden, thus enhancing the therapeutic efficacy and prolonging overall survival [32].

Pre-clinical and clinical studies of different kinds of HDACi combined with immune checkpoint inhibitor therapy have yielded some exciting results. For example, CT26 (colon) tumors or 4T1 (breast) tumors could not be eradicated in mice using anti-PD-L1 or anti-CTLA-4 therapy. Treating these tumors with antibodies targeting PD-L1 or CTLA-4 plus the HDACi Entinostat greatly improved the efficacy [33,34]. Furthermore, the anti-PD-L1 antibody Atezolizumab and the HDACi Entinostat are being tested in phase I/II clinical trials in patients with advanced triple negative breast cancer (aTNBC) to assess the efficacy of this combined therapy (NCT02708680). Similarly, the combination of Pembrolizumab and Vorinostat is under phase I/II investigation to assess its efficacy in patients with stage IV NSCLC (NCT02638090). 

(2) HDAC inhibitors combined with immune checkpoint inhibitors targeting CTLA-4 in pre-clinical and clinical trials 

CTLA-4 is another important immune checkpoint. The strategy of combing HDAC inhibitors with an antibody targeting CTLA-4 is another potential approach to treat cancer. Recent research indicated that HDAC inhibitors can enhance the efficacy of an antibody targeting CTLA-4 by modulating the relationship between immune and cancer cells, hence upregulating the recognition and elimination of tumor cells. Like CD28, CTLA4 is able to bind to CD80 and CD86, which are expressed on the surface of tumor cells and APCs [31]. Research has shown that HDACis can upregulate the expression of CD80 and CD86, hence improving tumor cell immunogenicity, promoting the elimination of tumor cells, and enhancing the treatment efficacy of antibodies targeting CTLA-4 [32]. Combination strategies involving HDAC inhibitors together with an antibody against CTLA-4 are still being investigated in various clinical trials. These include Ipilimumab combined with Entinostat in the treatment of Breast cancer (phase I, NCT02453620) and Ipilimumab combined with Panobiostat in the treatment of unresectable stage III/IV melanoma (NCT02032810). An overview of other HDACi and checkpoint inhibitor combinations is shown in Table 2.

#### 4.2.2. HDAC Inhibitors Combined with Other Immunotherapies

(1) HDAC inhibitors combined with OV (oncolytic virus) therapy

Recently, a prospective combination therapy was proposed using HDACis and oncolytic viruses (OVs) [7]. The application of OVs mainly relies on the possibility of identifying virus strains (or making mutants) which can infect tumor cells and replicate with very high efficiency, giving rise to a series of oncolytic cycles [7,29]. The HDACi will dampen the ‘natural’ antiviral response of the immune system, thus enhancing viral replication and promoting the lysis of tumor cells [7,34,40]. Recently, this combined therapy was further investigated in a clinical trial, which showed that OV together with an antibody that targets PD-1 was effective in treating melanoma [51].

(2) HDAC inhibitors combined with exosome therapy

Another promising immunotherapy strategy is the use of exosomes in cancer vaccines [7]. Exosomes are small membrane-enclosed sacs which are released from cells and play important roles in the storage and transport of molecular constituents, including proteins and RNAs. Previous work has shown that exosome-mediated transfer of these molecules from one cell to another modulates the immune response to pathogens and tumors [7]. Secretion of exosomes by tumor cells mediates immune suppression. After cancer cells were treated with HDACis, the extracted exosomes enhanced the efficacy of anti-tumor vaccines. Xiao et al. demonstrated that the HDAC inhibitor Entinostat was able to enhance the non-specific immune response of exosomes derived from HepG2 cells [52]. This could provide a potential tumor vaccine strategy against liver cancer. However, this strategy is still under investigation, and further studies are needed to gain a greater insight into this type of adjuvant therapy. 

### 4.3. Investigation of Histone Methylation Modification Inhibitors in Combination with Immunotherapies in Pre-Clinical and Clinical Trials

#### 4.3.1. Histone Methylation Modification Inhibitors in Combination with Immune Checkpoint Inhibitors in Pre-Clinical and Clinical Trials

(1) Histone methylation modification inhibitors combined with immune checkpoint inhibitors targeting PD-1/PD-L1 in pre-clinical and clinical trials

In general, immune checkpoint inhibitors are less commonly combined with HMTis or HDMis than with HDACis. A pre-clinical study of the HMTi CPI-1205, which targets EZH2, indicated that intratumoral regulatory T cells were reprogrammed, which enhanced the immune response against the cancer cells [42]. In addition, another study showed that the EZH2 inhibitors DZNep and EPZ6438 combined with 5-AZA, an inhibitor of DNA methyltransferase (DNMT), improved the therapeutic efficacy of the immune checkpoint inhibitor anti-CXCR3, which targets PD-L1 [53]. The above pre-clinical results suggest that the clinical usage of HMTis or HDMis in combination with immunotherapy drugs can overcome the limitations of immunotherapy. Some reports showed that, in some cancer line cells, repressive histone marks such as H3K27me3 were enriched in the promoters of genes encoding CTAs, hence allowing the cells to escape immune surveillance [32]. Inhibitors targeting EZH2 increase the expression of CTAs and enhance the immunogenicity of the tumor cells. In addition, a recent study indicated that knockdown of the demethylase KDM1 or KDM5B enhanced the DNMTi-mediated activation of CTA expression, suggesting that other HDMs or HMTs are also involved in regulating the expression of CTAs [11]. Clinical trials investigating the antibody Atezolizumab (MPDL3280A), which targets PD-L1, together with Tazemetostat, which inhibits EZH2, have been performed at phase I in relapsed or refractory follicular lymphoma and diffuse large B-cell lymphoma (NCT02220842). 

(2) Histone methylation modification inhibitors combined with immune checkpoint inhibitors targeting CTLA-4 in pre-clinical and clinical trials

As mentioned above, HMTs and HDMs play important roles in regulating the expression of TAAs on tumor cells, the induction of chemokines, and the expression of other immune components. Inhibitors designed to target these HMTs and HDMs can therefore modulate the expression of multiple factors involved in the immune response, hence enhancing the efficacy of immunotherapy. Goswami et al. reported that an inhibitor targeting EZH2 reduced the population of immunosuppressive cells and enhanced the therapeutic efficacy of an antibody targeting CTLA-4 [54]. The results support the promising potential of this combined approach. An overview of other combinations is shown in Table 2.

#### 4.3.2. Histone Methylation Modulation Inhibitors in Combination with Other Immunotherapies in Pre-Clinical and Clinical Trials

Inhibitors of histone methylation modification have also shown enhanced therapeutic efficacy in combination with adoptive cellular immunotherapy. Adoptive cell transfer (ACT) consists of purification, in vitro modification, expansion, and infusion back into patients of therapeutically-relevant immune cell types [11]. It has been shown that ACT promotes the infiltration of autologous TILs and extends the repressive effect in patients with metastatic melanoma [11]. The success of this therapy depends on the recognition of antigen expression in cancer cells by the engineered T cells. Previous research indicated that histone methylation inhibitors combined with ACT therapy enhance the expression of antigens on the surface of tumor cells, hence improving the therapeutic efficacy [11]. Peng et al. proposed that the selective inhibitor GSK126, which targets EZH2, improved the efficacy of adaptive T cell transfusion therapy in ovarian cancer-bearing mice [53]. 

## 5. Conclusions

Histone modifications are common epigenetic changes in eukaryotes, and the dysregulation of histone modification enzymes is related to the onset and development of various types of cancer. However, the mechanisms underlying histone modifications and their role in the onset and development of cancer are still not fully understood. Several agents that target histone modification enzymes have been approved by the FDA. Although some histone modification inhibitors have achieved satisfactory clinical efficacy, some limitations, such as patient non-responsiveness and relapse, have been observed. Some reports have shown that histone modification inhibitors modulate the immune response, and this implies that combining such inhibitors with immunotherapy may result in greater efficacy. Currently, histone modification inhibitors are being used in clinics together with immunotherapy agents, thus establishing a new direction for tumor therapy. Several histone modification inhibitors and immune checkpoint blockade drugs have been approved by the FDA or the European Medicines Agency (EMA) for the treatment of various cancers. Pre-clinical studies are uncovering the underlying mechanisms by which histone modification inhibitors modulate the immune system. The combination of histone modification inhibitors with immune checkpoint inhibitors may deliver long-lasting and efficacious therapeutic effects by affecting many different aspects of the response. Some pre-clinical and clinical results have already shown that this combined therapy overcomes the limitations of monotherapy. However, the mechanisms by which the combined agents modulate the immune response still need further investigation. Reducing the toxicity of combination therapy and the adverse events caused by the non-specific HATis, HDACis, and HMTis remains a great challenge. It is also necessary to further understand how histone modifications affect immune function and how the immune checkpoint modulates the occurrence and development of tumors. Improvements in therapy schemes are also needed. This work is likely to provide the basis of a promising approach to cancer therapy, and the bio-markers discovered may be used to assess and monitor the response of cancer patients during treatment.

## Figures and Tables

**Figure 1 genes-09-00633-f001:**
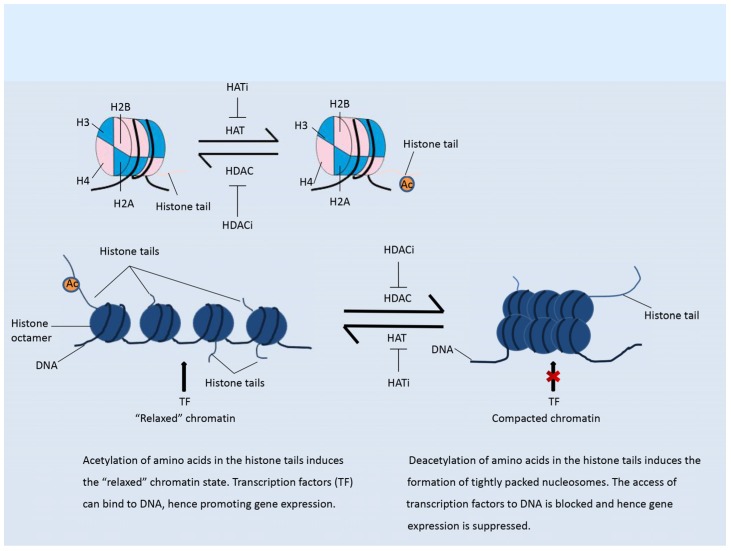
Histone acetylation. **Left panel**: Acetylation of histones maintains the “relaxed” status of chromatin. Transcription factors (TF) can easily bind to the DNA, hence promoting gene expression **Right panel**: Deacetylation of histones induces the formation of tightly packed nucleosomes, which suppress the binding of transcription factors to DNA and hence inhibits gene expression.

**Figure 2 genes-09-00633-f002:**
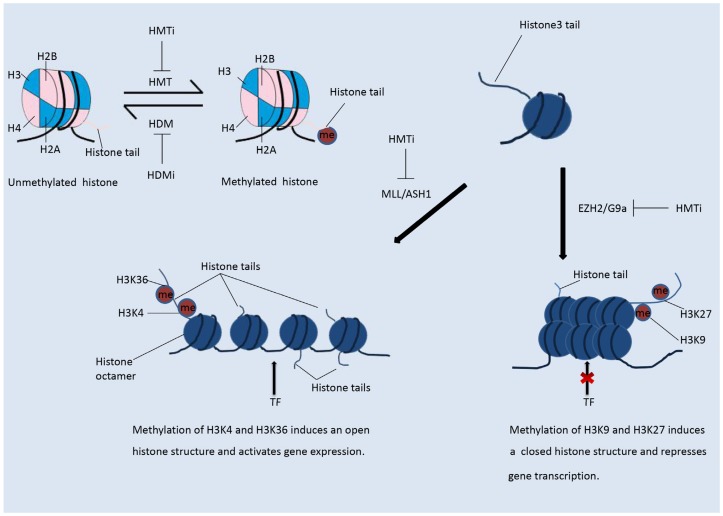
Histone methylation. The methylation status of a histone is reversible. In contrast to histone acetylation, methylation of histones on different residues suppresses or enhances gene transcription. Methylation of H3K4 and H3K36 induces an open histone structure and hence promotes transcription. Methylation of H3K9 and H3K27 induces a compacted histone structure and hence suppresses gene transcription.

**Figure 3 genes-09-00633-f003:**
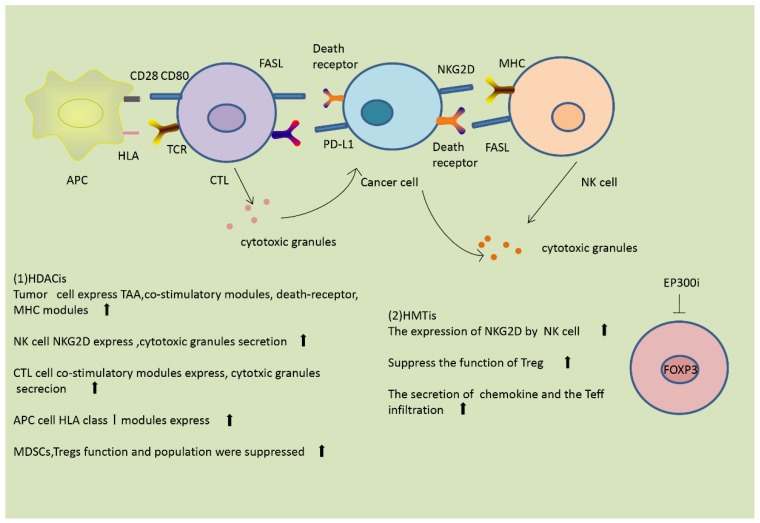
Histone modifications modulate the immune response. (1) HDACis are able to modulate the immune innate and adaptive immune host cells and other components of the immune system. HDACis enhance the expression of tumor-associated antigens (TAAs), co-stimulatory molecules, death receptors, and major histocompatibility complex (MHC) molecules on the surface of tumor cells. In natural killer (NK) cells, HDACis enhance the expression of NKG2D and the secretion of cytotoxic granules. In antigen-presenting cells (APCs), HDACis induce the expression of human leukocyte antigens (HLA) class I molecules. In cytotoxic T lymphocytes (CTLs), HDACis increase the expression of co-stimulatory molecules and the secretion of cytotoxic granules. HDACis also suppress the function and populations of myeloid-derived suppressor cells (MDSCs) and Tregs. (2) Histone methyltransferase inhibitors (HMTis) enhance the expression of NKG2D and promote NK cell activation. HMTis suppress the function of Tregs, enhance the secretion of chemokines, and promote infiltration of Teffs.

**Table 1 genes-09-00633-t001:** The structure of histone modification inhibitors.

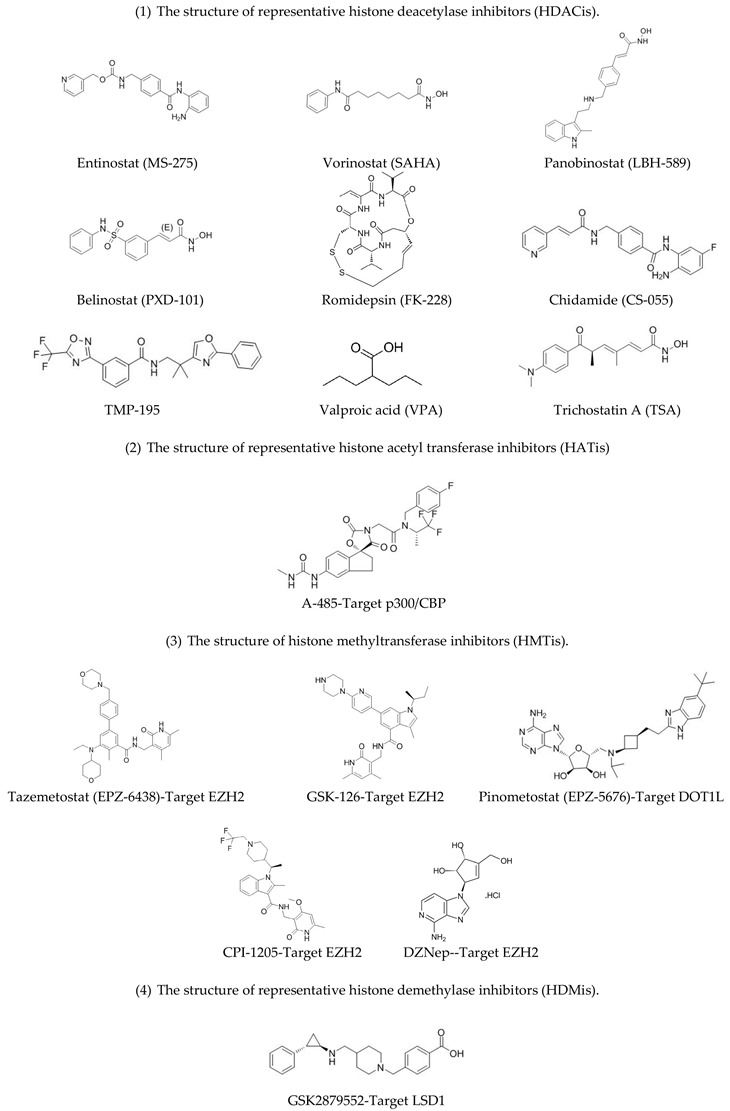

**Table 2 genes-09-00633-t002:** Current clinical trials of histone modification inhibitors combined with immunotherapy agents.

Identifier	Recruitment Status	Phase	Cancer Type	Immune Checkpoint Inhibitors	Epigenetic Drugs
NCT02437136	Recruiting	I/II	NSCLC and melanoma	Pembrolizumab	Entinostat
NCT03179930	Recruiting	II	Lymphoma Relapsed Refractory	Pembrolizumab	Entinostat
NCT02909452	Active, not recruiting	I	Advanced solid tumors	Pembrolizumab	Entinostat
NCT02395627	Recruiting	II	Breast Neoplasms	Pembrolizumab	Vorinostat
NCT02619253	Recruiting	I	Renal Cell Carcinoma, Urinary Bladder Neoplasms	Pembrolizumab	Vorinostat
NCT02538510	Active, not recruiting	I/II	Recurrent unresectable/metastatic HNSCC and SGC	Pembrolizumab	Vorinostat
NCT02512172	Recruiting	I	Colorectal Cancer	Pembrolizumab	Romidepsin with or without azacytidine(DNMTi)
NCT02697630	Recruiting	II	Metastatic Uveal Melanoma	Pembrolizumab	Entinostat
NCT02453620	Recruiting	I	Breast Adenocarcinoma HER2/Neu Negative Invasive Breast Carcinoma	Ipilimumab	Entinostat
NCT03552380	Recruiting	II	Renal Cell Carcinoma	Pembrolizumab plus Ipilimumab	Entinostat
NCT03250273	Recruiting	II	Cholangiocarcinoma and Pancreatic Cancer and Metastatic Pancreatic Cancer	Nivolumab	Entinostat
NCT03278782	Recruiting	I/II	Lymphoid Haematopoietic Malignant Neoplasms Cutaneous T-Cell Lymphoma Refractory Cutaneous T-cell Lymphoma	Pembrolizumab	Romidepsin
NCT03150329	Recruiting	I	Relapsed or Refractory Diffuse Large B-Cell Lymphoma, Follicular Lymphoma, or Hodgkin Lymphoma	Pembrolizumab	Vorinostat
NCT02915523	Active, not recruiting	I/II	Epithelial Ovarian Cancer, Peritoneal Cancer, Fallopian Tube Cancer	Avelumab	Entinostat
NCT02708680	Recruiting	I/II	Breast Cancer	Atezolizumab	Entinostat
NCT02638090	Recruiting	I/II	Lung Cancer, Non-small Cell Lung Cancer	Pembrolizumab	Vorinostat
NCT02220842	Active, not recruiting	I	Lymphoma	Atezolizumab administered with Obinutuzumab	Tazemetostat
NCT03525795	Recruiting	I/II	Advanced Solid Tumors	Ipilimumab	CPI-1205
NCT02032810	Active, not recruiting	I	Melanoma/Skin Cancer	Ipilimumab	Panobinostat

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
