# Peer review of "Enhancing the Anticancer Efficacy of Immunotherapy through Combination with Histone Modification Inhibitors"

_genes, 2018, doi:10.3390/genes9120633_

Round 1

Reviewer 1 Report

In this review manuscript, Sun et. al. focused on two emerging fields---epigenetics and immunotherapy. While the reviewer articles related to the two fields individually are abundant, the work at the interface is limited. This manuscript nicely covered the gap. The manuscript is organized by short reviews of epigenetic modulators and immunotherapy that have clinical potentials and then mainly focus on the status of their combination treatment. The manuscript is well organized, rationally presented and well written. I suggest publishing this work after addressing the following minor concerns and typos:

(1)    Through the text, “P53” should be “p53”.

(2)    Through the text, “P300” should be “p300”.

(3)    While there is no inhibitor of HATs in the clinical trail, the newly reported p300 inhibitor showed such potential (Nature, 2017, 550, 7674, 128). Given that it is the only compound and the best candidate against p300, some discussion should be included.

(4)    Through the text, a handful of small-molecule epigenetic modulators were discussed. It will be helpful to include a table of the representative compounds with their chemical structures.

Full names for some abbreviations such as APCs were missed. 

Reviewer 2 Report

The manuscript submitted by W. Cui / L. Wang and co-workers aims to summarize the current knowledge regarding combined anticancer therapy – immune checkpoint and histone modification inhibitors based.

The paper is very interesting, well written and focused, with the following suggestions I am convinced that the current manuscript will be improved and readability will be ensured.

-          Figure 1 quality should be improve as in the upper panel we see that the nucleosomes are built form 4 not 8 histone proteins and the scheme suggests the histone tail come from the DNA not from the protein. (the same in mistake is Figure 2 – top left picture)

-          Figure 1 the capture – histone modifications may cause open / closed chromatin structure not histone structure

-          Table 1. - The review base on the histone modification inhibitors, whereas in the table we can find the examples combined therapy with azacitidine. It should be indicated that it a DNMTs inhibitor or removed.

-          The references are a bit in a mess and should be formatted regarding the journal guidelines.

Overall I do recommend the publication of the paper in the Genes journal.

Author Response

1. <Comments>

: Figure 1 quality should be improve as in the upper panel we see that the nucleosomes are built form 4 not 8 histone proteins and the scheme suggests the histone tail come from the DNA not from the protein. (the same in mistake is Figure 2 – top left picture)

<Answer>

: According to the reviewer’s comment, we amended the relevant part in Figure 1 and Figure 2 respectively.

®

2. <Comments>

: Figure 1 the capture – histone modifications may cause open / closed chromatin structure not histone structure

<Answer>

: According to the reviewer’s comment, we amended the relevant part in Figure 1 to make readers clearly understand histone modifications may cause open/closed chromatin structure .

®

3. <Comments>

: Table 1. - The review base on the histone modification inhibitors, whereas in the table we can find the examples combined therapy with azacitidine. It should be indicated that it a DNMTs inhibitor or removed.

<Answer>

: According to the reviewer’s comment, we removed some examples combined therapy with azacitidine and added some examples combined therapy with histone modulators. And, we indicated azacytidine a DNMTi in the indentifier NCT02512172 clinical trail example.

®

Identifier  Recruitment     Phase   Cancer type         Immune checkpoint      Epigenetic drugs

            status                                   inhibitor

NCT02437136     Recruiting       I/II       NSCLC and melanoma              Pembrolizumab                 Entinostat

NCT03179930     Recruiting        II       Lymphoma Relapsed Refractory      Pembrolizumab                 Entinostat

NCT02909452     Active, not        I       Advanced solid tumors               Pembrolizumab                 Entinostat

                  recruiting

NCT02395627     Recruiting        II      Breast Neoplasms                   Pembrolizumab                 Vorinostat

NCT02619253     Recruiting        I       Renal Cell Carcinoma, Urinary        Pembrolizumab                 Vorinostat

                                            Bladder Neoplasms

NCT02538510     Active, not       I/II     Recurrent unresectable/metastatic      Pembrolizumab                 Vorinostat

recruiting                    HNSCC and SGC

NCT02512172     Recruiting        I        Colorectal Cancer                      Pembrolizumab            Romidepsin with or without

                                                                                                         azacytidine(DNMTi)                  

NCT02697630     Recruiting        II       Metastatic Uveal Melanoma          Pembrolizumab                  Entinostat

NCT02453620     Recruiting        I       Breast Adenocarcinoma HER2/Neu      Ipilimumab                     Entinostat

                                          Negative   Invasive Breast Carcinoma

NCT03552380     Recruiting        II        Renal Cell Carcinoma             Pembrolizumab plus               Entinostat

Ipilimumab

NCT03250273     Recruiting        II        Cholangiocarcinoma and Pancreatic    Nivolumab                    Entinostat

                                            Cancer   and Metastatic Pancreatic

Cancer

NCT03278782    Recruiting        I/II    Lymphoid    Haematopoietic Malignant       Pembrolizumab                 Romidepsin

                                           Neoplasms Cutaneous T-Cell    Lymphoma 

                                           Refractory Cutaneous  T-cell   Lymphoma                       

NCT03150329    Recruiting         I       Relapsed or Refractory Diffuse Large       Pembrolizumab                Vorinostat

B-Cell Lymphoma, Follicular Lymphoma,

 or Hodgkin   Lymphoma                                                             

NCT02915523     Active, not       I/II      Epithelial Ovarian Cancer, Peritoneal         Avelumab                    Entinostat

                  recruiting                 Cancer, Fallopian Tube   Cancer 

NCT02708680     Recruiting        I/II        Breast  Cancer                      Atezolizumab                   Entinostat

NCT02638090     Recruiting        I/II        Lung    Cancer                         Pembrolizumab                 Vorinostat

Non-small Cell Lung Cancer

NCT02220842     Active, not         I         Lymphoma                        Atezolizumab   administered      Tazemetostat  

                  recruiting                                                         with Obinutuzumab      

NCT03525795     Recruiting        I/II       Advanced   Solid Tumors                   Ipilimumab                   CPI-1205

NCT02032810     Active, not        I         Melanoma/Skin Cancer                Ipilimumab                   Panobinostat

                  recruiting    

4. <Comments>

: The references are a bit in a mess and should be formatted regarding the journal guidelines.

<Answer>

: According to the reviewers comment, we check up our manuscript seriously and corrected formats of these references on the basis of journal guidelines. Futhermore, we adjusted these reference and added this reference 

[22]Lasko, L. M., Jakob, C. G., Edalji, R. P.Discovery of a selective catalytic p300/CBP inhibitor that targets lineage-specific tumours.[J]. Nature,2017,550(7674):128-132.
